# Transcriptional Differences for COVID-19 Disease Map Genes between Males and Females Indicate a Different Basal Immunophenotype Relevant to the Disease

**DOI:** 10.3390/genes11121447

**Published:** 2020-12-01

**Authors:** Tianyuan Liu, Leandro Balzano-Nogueira, Ana Lleo, Ana Conesa

**Affiliations:** 1Microbiology and Cell Science, Institute for Food and Agricultural Research, University of Florida, Gainesville, FL 32611, USA; tianyuan.liu@ufl.edu (T.L.); leobalzano@ufl.edu (L.B.-N.); 2Internal Medicine and Hepatology, Humanitas Clinical and Research Center-IRCCS, Department of Biomedical Sciences, Humanitas University, MI 20089 Rozzano, Italy; ana.lleo@humanitas.it; 3Genetics Institute, University of Florida, Gainesville, FL 32608, USA

**Keywords:** COVID-19, sex, age, DeCovid app, basal immunophenotype

## Abstract

Worldwide COVID-19 epidemiology data indicate differences in disease incidence amongst sex and gender demographic groups. Specifically, male patients are at a higher death risk than female patients, and the older population is significantly more affected than young individuals. Whether this difference is a consequence of a pre-existing differential response to the virus, has not been studied in detail. We created DeCovid, an R shiny app that combines gene expression (GE) data of different human tissue from the Genotype-Tissue Expression (GTEx) project along with the COVID-19 Disease Map and COVID-19 related pathways gene collections to explore basal GE differences across healthy demographic groups. We used this app to study differential gene expression of COVID-19 associated genes in different age and sex groups. We identified that healthy women show higher expression-levels of interferon genes. Conversely, healthy men exhibit higher levels of proinflammatory cytokines. Additionally, young people present a stronger complement system and maintain a high level of matrix metalloproteases than older adults. Our data suggest the existence of different basal immunophenotypes amongst different demographic groups, which are relevant to COVID-19 progression and may contribute to explaining sex and age biases in disease severity. The DeCovid app is an effective and easy to use tool for exploring the GE levels relevant to COVID-19 across demographic groups and tissues.

## 1. Introduction

Severe acute respiratory syndrome coronavirus 2 (SARS-CoV-2), the virus that causes coronavirus disease 2019 (COVID-19), is without a doubt the most severe health, economic and social threat of our time. COVID-19 has followed two major infection waves in spring and fall 2020 reaching one million deceased people by September 2020. Our understanding of the disease has evolved over time. While initially COVID-19 was considered to be a severe atypical pneumonia, we have now learned that the virus can infect many different tissues. Moreover, the disease affects blood coagulation and can cause an exacerbated immune response referred to as the “cytokine storm”. Although clinical management of COVID-19 has rapidly evolved over time, no curative treatment is available, and current recommendations support the use of steroids, oxygen and a prophylactic dose of heparin. While accelerated vaccine development efforts have resulted in tremendous advances in a relatively short time, the WHO acknowledges that a vaccine solution for the general population might not be ready before mid to late 2021. At the same time, very recent reinfection cases suggest that immunity may not always persist, and active disease surveillance of past cases may be required. As the second wave of the pandemic steadily progresses in Europe and the US, possibly with new variants of the virus that affect a wider range of the population, novel or improved treatment targets and options are likely to remain important for the management of this disease.

One of the most intriguing aspects of COVID-19 is the varying grades of severity in which it affects different people. Although acknowledged risk factors include pre-existing conditions such as cardiovascular disease [1,2], diabetes [3,4], obesity [5,6], age [7,8] and sex [8,9,10] and suggest a relationship between physical condition and disease progression, the precise pathways of this relationship have not been clearly established yet. Sex-associated differences between males and females that have been proposed to affect COVID-19 incidence include lifestyle (e.g., smoking and drinking) and mental health. To exemplify, the smoking population of males is higher than females, and smoking may be a risk factor for severe COVID-19 [11,12]. In contrast, men differ from women during the pandemic in sleeping patterns and signs of depression [13]. Furthermore, a recent study showed that male and female COVID-19 patients differed in their immune response, with the former showing a stronger cytokine response while the latter has a higher T-cell activation pattern [14]. Moreover, the majority of the death cases are among the elder, implying there is a significant age bias [15]. Interestingly, the pattern of COVID-19 risk factors is not fully shared with other similar recent pandemics such as SARS, MERS or H1N1. For example, SARS patients were more frequently healthy young people [16]. Age and complications, but not gender, were the most significant risk factors for mortality in the Arab and South Korean Studies of MERS [17]. Several studies on the 2009 H1N1 influenza revealed that younger age, chronic conditions and female sex, were risk factors for the disease [18]. This suggests that the observed COVID-19 risk factors, especially those associated with differences at age and sex groups, might be the result of specific interactions between SARS-CoV-2 and intrinsic physiological characteristics, including the basal immune system characteristics of these population groups. However, which specific aspects of the sex and age related immunophenotype imply a pre-existing condition and how this relates to COVID-19 remains to be explored.

The COVID-19 Disease Map initiative was launched in May 2020 with the aim of creating a collection of genes and pathways relevant to SARS-CoV-2 infection derived from the growing COVID-19 literature. This resource contains over 800 COVID-19 genes that have been associated to the disease, and can be used to understand the SARS-CoV-2 virus–host interaction, the SARS-CoV-2 replication cycle and the adaptive and innate immune response pathway [19]. Additionally, WikiPathways COVID-19 related pathways genes are also available on the same website (https://covid.pages.uni.lu/) describing genes related to the COVID-19 disease process, including cytokines and interferon, together with the pathways for the life cycle of SARS-CoV and MERS-CoV. The COVID-19 Disease Map and WikiPathways COVID-19 related pathways are valuable resources to investigate the molecular responses to SARS2-CoV-2 infection and to understand the biological pathways leading to severe manifestations of the disease. These databases also create an opportunity for interrogating existing molecular data on differences associated with COVID-19 risk factors in the general population.

This paper presents the DeCovid, a Shiny app for exploring basal expression level differences between two demographic risk factors, age and sex, for the COVID-19 Disease Map and COVID-19 related pathway genes expressed at several human tissues. DeCovid allows us to interrogate COVID-19 Disease Map and COVID-19 related pathway genes globally or individually. We used data from the GTEx database, which contains RNA-seq profiles for hundreds of demographically diverse healthy individuals in multiple human tissues and cell types. This resource was used to study basal expression differences in COVID-19 associated genes between different demographic groups (e.g., men versus women, and young people vs. old people) to understand the baseline immune patterns they present towards COVID-19. We found that a similar immunological state is prevalent in young people and women, which is different than that found in men and old people. Interestingly, some of these differential pathways have been shown to be relevant for disease severity progression and are characteristic of the sex-biased immune response to the virus, providing ground for hypotheses on the molecular basis of the COVID-19 sex and age bias. We anticipate that the DeCovid app will be a useful tool for researchers to explore the molecular etiology of COVID-19 demographic differences.

## 2. Material and Methods

### 2.1. Datasets

We used RNA-seq data from the Genotype-Tissue Expression project (GTEx) containing data from 44 tissue sources obtained from individuals from both sexes who are deceased but were considered healthy, and that belonged to a wide range of ages [20]. Gene read counts matrix and annotation files were obtained from the GTEx portal (https://www.gtexportal.org/home/datasets). GTEx data were analyzed for possible biases in relation to the indicated cause of death using the principal component analysis, and blood samples belonging to individuals with a “ventilator case” were discarded (Appendix A). The COVID-19 Disease Map genes were downloaded from https://git-r3lab.uni.lu/covid/models, on 5 May 2020, while the list of genes in COVID-19 related pathways was obtained from https://github.com/wikipathways/cord-19 on 17 March 2020. The COVID-19 Disease Map genes is a database of manually curated genes extracted from papers related to the SARS-CoV-2 immune responses and replication cycle [19]. The list of genes in COVID-19 related pathways was created by the WikiPathways group using machine learning to mine 9996 PMC papers present in the COVID-19 Open Research Dataset (CORD-19).

### 2.2. DeCovid Software

The DeCovid software is a Shiny app written in R with a user-friendly interface. The app can be installed through a Docker image or directly downloaded from GitHub (see Appendix A). DeCovid already has all essential data from GTEx integrated, and no additional downloads are necessary. Differential gene expression is calculated using *edgeR* [21] and multiple testing correction is applied following the Benjamini and Hochberg (BH) method [22]. Results are provided either as lists of differentially expressed genes, heatmaps of sex and age mean expression values and gene-specific boxplots showing the distribution of expression values across demographic groups. GO enrichment analysis of significant gene sets is provided through the clusterprofile R package and uses the list of the COVID-19 Disease Map as a reference set [23]. The tool is available at https://github.com/ConesaLab/DeCovid.

## 3. Results

### 3.1. DeCovid as a User-Friendly Tool to Explore Gene Expression of COVID-19 Related Genes in Healthy Individuals 

The DeCovid shiny app combines GTEx data from a selection of relevant human tissues and cell types with the COVID-19 Disease Map database and COVID-19 related pathways to facilitate the analysis of basal immune differences in the healthy human population for genes described to be important for COVID-19. We included data from whole lungs, heart, kidney, stomach and brain as they were tissues described to be affected by SARS-CoV-2 infection [24]. We also included blood, spleen and lymphocytes since they are critical components of the human immune system. Being a fundamental tissue for the study of basal immunity, blood contains a myriad of innate and adaptive immune cells and is connected to different immune organs [25], while spleen and lymphocytes are crucial parts of the lymphatic system, playing a significant role in immunity [26]. The sample size for different tissues is different in the GTEx datasets. We excluded samples that showed biased expression after PCA quality control (Appendix A). The largest sample size included in DeCovid is whole blood with 330 samples, while the smallest dataset is kidney-Cortex with 85 samples (Figure 1a). The number of COVID-19 Disease Map genes was 802, while COVID-19 related pathways involved 1523 genes. The total number of unique genes in the two datasets was 2087. The gene ontology (GO) term enrichment analysis indicates that genes related to cytokines, response to virus, innate immune response, transcriptional regulation, metabolic process and kinase activity populate this collection (Figure 1b).

The DeCovid app offers a user-friendly interface for the analysis of these data by researchers without strong bioinformatics skills (Figure 2). Extensive documentation and video tutorials are provided for a quick start. Users should indicate the demographic factor for differential expression analysis. If age is selected, the age threshold value to classify samples as old or young should be specified. The user may also select specific control and experimental groups (i.e., a certain age range) to perform custom analysis instead of using the whole dataset. Once a significance *p*-value and a fold-change threshold are provided, the app computes differential expression and provides results as a list of genes and heatmaps with sex and age groups mean values (Figure 2). Additionally, users can explore the distribution of gene expression values for specific genes of the COVID-19 Disease Map collection or investigate the enrichment of specific immune response-related functions among genes with sex or age expression biases.

### 3.2. DeCovid Shows Wide-Spread Gene Expression Differences in COVID-19 Disease Map Genes in Sex and Age Groups at Diverse Tissues and Cell Types

To explore the utility of DeCovid, we analyzed the extent of gene expression differences between different demographic groups and across different tissues and cell types. We performed this analysis on tissues that are relevant to SARS-CoV-2 infection. The analysis revealed a wide range of significant gene expression differences for COVID-19 genes when comparing sex and age groups for all evaluated samples (Table 1). Differences were lower when imposing a fold-change threshold of 0.5 for the magnitude of the mean expression level. Whole-blood and spleen were the tissues that presented the highest number of expression differences for COVID-19 genes. From this simple analysis, we concluded that the COVID-19 Disease Map represents a disease signature with significant differences across demographic groups that may be worth exploring to hypothesize on the molecular basis for differences in disease incidence among these groups.

### 3.3. DeCovid Identified Shared and Specific Immune Gene Expression Signatures for Demographic Groups with Lower Risk for COVID-19

To further understand differences in the immunophenotype between demographic groups that might relate to their differential susceptibility to the severe forms of COVID-19, we analyzed in detail gene expression differences in three tissues relevant for systemic immunity: blood, spleen and lymphocytes. Although each contrast (sex or age) revealed many specific differentially expressed genes across different tissues/cells (see Appendix A for a complete list of differentially expressed genes), we found a number of interesting commonalities that indicate a shared immunophenotype for lower risk groups that are worth discussing (Figure 3, Appendix A). We found high expression of genes involved in the innate immune response across different tissues/cells in both young people and females. These highly expressed genes involved complement system genes such as C2, C3, C9 and CRP. Additionally, young people showed a higher expression level of matrix metalloproteinases genes, which are inflammatory regulators, though females do not show this pattern. In contrast, females present a higher level of interferon expression, which is not found in young people. On the other hand, males showed higher basal expression of genes encoding for proinflammatory chemokines (CXCL14, CXCL2 and CXCL8) and cytokines (IL2, IL3 and IL22). These gene expression differences point to four major immune response pathways differently active in men and women, young and old: interferon-mediated antiviral responses, complement-mediated cell lysis, matrix metalloproteinases regulation of inflammatory and the cytokine activation response. Interestingly these four processes have been indicated as distinctive pathways in the severity of the SARS-CoV-2 response [27,28,29].

#### 3.3.1. Complement System

Young people showed high expression of complement system genes in different immune tissues (Figure 3), women exhibited higher expression in whole blood and lymphocytes. The complement system is the first line of the innate immune system, which detects virus-infected cells at early stages and lyses them to prevent further infection [30], and coagulation factors can boost the complement system through the cleavage of C3 and C5, resulting in activation [31]. Previous research has shown that in the early stages (first week) of the infection, the complement system plays a positive role in controlling SARS-CoV-2 infection [32]. Additionally, clinical COVID19 data suggest that the decrease of C3, a key player in this pathway, is associated with a higher probability of death [33,34]. Nonetheless, a high expression of genes related to the complement system is not always beneficial, especially for severe COVID-19 patients, as it could also lead acute respiratory distress syndrome (ARDS), and eventually respiratory failure [35,36,37]. Therefore, we speculate that a strong basal complement system in young people and women may favor a faster response in these groups to viral infection resulting in lysis of the infected cells, thereby helping to control SARS-CoV-2 at an early infection stage. Figure 4 illustrates this model by depicting the upregulation of the complement system pathway in the young versus old male population.

#### 3.3.2. Type 1 Interferon (IFN) Family

Data showed that females exhibit higher levels of anti-viral type 1 interferons IFNA17, IFNA2, IFIT1, IFIT3 and IFNE in the immune tissues/cells, which serve as the first line of defense against viral and bacterial infections [38] (Figure 5). Moreover, type 1 IFN enhances T-cell survival and effector functions through initial antigen recognition [39], which may facilitate restricted virus proliferation. This transcriptional expression profile is in agreement with recent clinical studies showing that female patients have high T-cell levels [14], which were postulated as a critical factor for the differential fatality incidence between sexes. Therefore, with a generally higher-level expression of type 1 interferons, females are more likely to maintain a high T-cell level in the early stages of the infection, which might lead to a milder evolution of the disease.

#### 3.3.3. Matrix Metalloproteases

Data showed that young people have a higher expression of matrix metalloproteases (MMPs) genes in all immune tissues/cells (Figure 3). In addition, whole blood samples from older females presented high expression of MMP1 and MMP19 genes. MMPs play a significant role in the anti-inflammatory regulatory process, especially acute inflammation [40]. For example, M2, the anti-inflammatory macrophages, are regulated by MMP10, MMP19 and MMP28 and prevent acute inflammation [41]. In animal models, MMP3 showed an anti-inflammatory effect mediated by the serum-derived hyaluronan-associated protein (SHAP) [42], and MMP12 has been shown to turn off inflammatory signaling and stimulate anti-inflammatory processes [43]. Moreover, MMPs are involved in tissue repair, as MMP7 and MMP9 have been reported to participate in epithelial tissue repair after damage caused by acute respiratory distress syndrome (ARDS) [44]. Therefore, a higher baseline level of MMPs genes in young people and old females may reduce the risk for acute inflammation and ARDS when infected with SARS-CoV-2. However, high MMPs may not always play a beneficial role as they could also induce some inflammatory processes, which is why inhibition of MMPs has been proposed as therapy for severe patients of SARS-CoV-2 [45]. As for the complement system, we hypothesized that MMPs have a dual significance in COVID-19, being protective when at a higher level in the healthy population but harmful once the infection has been established.

#### 3.3.4. Cytokines and Chemokines

Expression of proinflammatory cytokines such as IL17, IL22 and IL3 were upregulated in older males, when compared to females (Figure 2). In addition, levels of chemokines CXCL2, CXCL8, CXCL20 and CXCL14 were higher in men, which supports the hypothesis that the first stage of male response against a viral infection is typically related to inflammation and Th17 cell differentiation processes. Death in COVID-19 positive patients results from the "cytokine storm" produced by the human immune system [29]. With a higher baseline of different cytokines and chemokines, we hypothesized that males are more likely to have a "cytokine storm" as a response to the infection. Figure 6 illustrates the different activation of proinflammatory cytokines between men and women for the spleen tissue.

## 4. Discussion

Here we present the DeCovid app as a resource to explore sex and age differential expression patterns in the healthy population for genes described to be involved in COVID-19 disease pathways. The GTEx data, used in this work, has been recently mined for sex-specific expression differences concluding that these are tissue-specific and generally small, which was also observed in our analyses [46]. This study focused on the sex-specific genetic effects on gene expression associated with complex genetic traits. In our application, we repurpose the valuable GTEx database for the study of COVID-19 relevant pathways while providing mechanistic models to interpret differential expression results.

We used DeCovid to investigate gene expression differences between men and women and between old and young people, with the aim of shedding light into possible basal differences in the characteristic immunophenotype of these demographic factors that could influence their distinct susceptibility for COVID-19. We found that four major immune response pathways were differentially expressed across different demographic groups in the healthy population (Figure 7). Women showed higher expression of the interferon genes, while men have higher proinflammatory Th17 cytokine genes. These two pathways have been proposed to be critical factors for the fatal response to the virus infection, with patients that present an interferon-mediated response having a better prognosis than those responding with a massive cytokine activation [28,47]. In agreement, animal models of SARS-CoV-1 and MERS-CoV infection showed that failure to elicit an early type 1 interferon response correlates with the disease severity [48]. Perhaps more importantly, these models demonstrate that timing is key, as type 1 interferons are protective at the early stages of a viral disease. On the other hand, clinical studies showed a difference in the immunophenotype between female and male COVID-19 patients with women having a more robust T-cell activation while male patients present a higher plasma level of proinflammatory cytokines [14]. Additionally, young people present a higher basal level of matrix metalloproteases than old people, which is helpful to regulate acute inflammation [40]. Notably, both young people and females have higher expression of genes associated to the complement system, an element of the innate immune system, which could help kill infected cells in the early stage of the infection, shown to be important for COVID-19 [30,32,33,34]. Interestingly, we found that females present similarities to young people in these innate immune system pathways. Altogether here we showed that multiple immune system differences, critical to COVID-19 progression, represent a sex-associated and age-associated pre-existing condition that is likely to create a differential predisposition to disease outcome in different demographic groups. Future studies should address whether these gene expression patterns translate into functional differences in patients infected with SARS-CoV-2, whether they have prognosis value and if they present any relationship with hormonal levels, which have also been linked to COVID-19 severity [49,50].

## 5. Conclusions

We have developed the DeCovid R shiny app to analyze baseline differences between sex an age demographic groups in the expression of COVID-19 relevant genes. Using this resource, we found several immune system related pathways, including interferon, complement system, matrix metalloproteinases, and cytokines that showed differential expression levels between men and women and/or between young and old people. We hypothesize that these population groups are characterized by distinct immunophenotypes that could contribute to their different patterns of progression to severe forms of the disease. Our results show that DeCovid is a useful resource to explore the expression of COVID-19 relevant genes in the healthy population.

## Figures and Tables

**Figure 1 genes-11-01447-f001:**
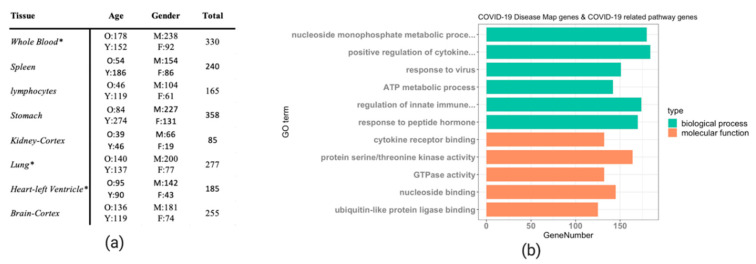
(**a**) Summary of the GTEx samples implemented the DeCovid app. O: old age (age ≥ 60); Y: young age (age < 60); M: males, F: females. * indicates that only a subpart of the original GETx data was used due to quality control issues. (**b**) Gene ontology enrichment analysis of the 1523 COVID-19 related pathways genes and 802 COVID-19 Disease Map genes. The most enriched GO terms by category are shown.

**Figure 2 genes-11-01447-f002:**
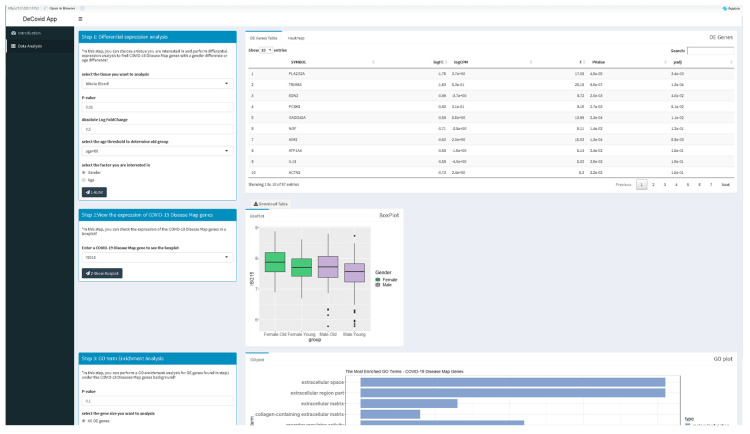
Screenshot of the DeCovid app showing the left dialog panels for input parameters (contrast type and significance thresholds), and on the right results panels with a list of differentially expressed genes, gene-specific expression plots and enrichment analysis.

**Figure 3 genes-11-01447-f003:**
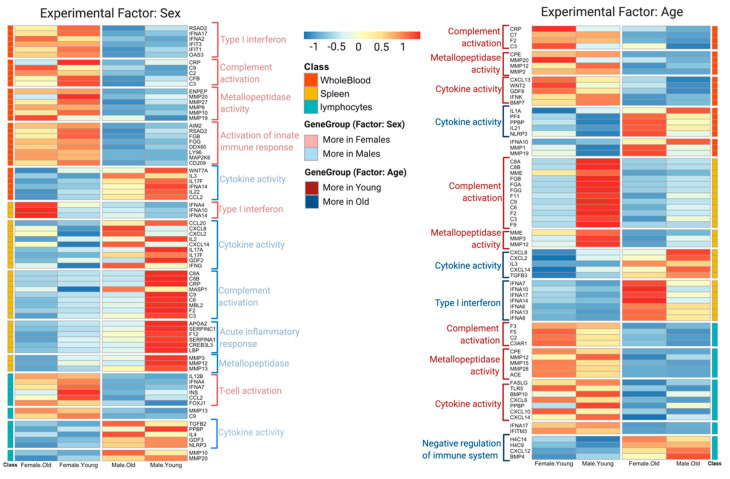
Heatmap with significant sex (**left**) and age (**right**) expression differences for selected COVID-19 genes and pathways. Results for different immune system tissues and cell types and shown.

**Figure 4 genes-11-01447-f004:**
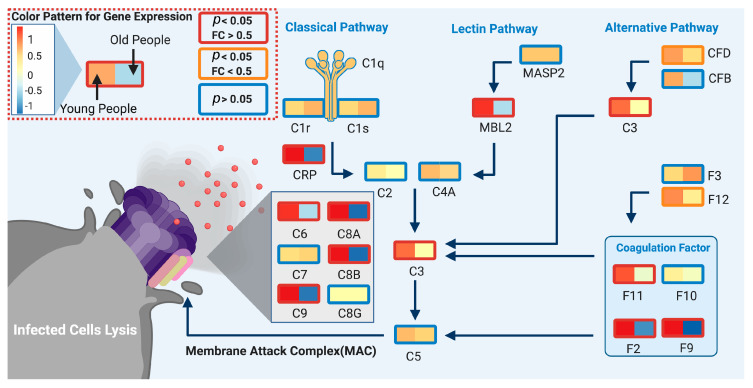
Differential expression of spleen complement system genes between male age groups. Boxes represent the mean gene expression value for the young (<60 years old; left) and the old (>60 years old; right) groups at spleen samples. Box line color indicates the significance level, red: *p* < 0.05 and fold change (FC) > 0.5; orange: *p* < 0.05 and FC > 0.5 and blue: *p* > 0.05 (not significant). The biological model indicates that multiple complement system and coagulation factor genes are more expressed in the younger than older male populations. Created with biorender.com.

**Figure 5 genes-11-01447-f005:**
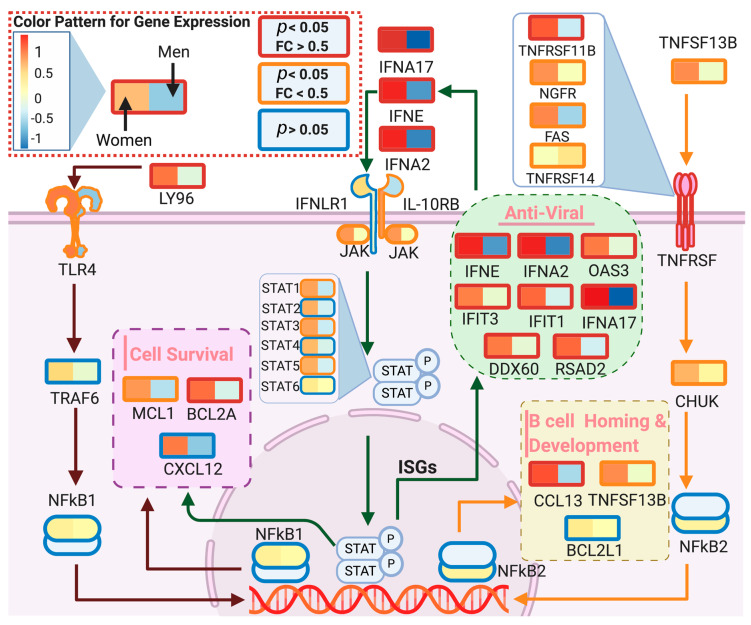
Differential expression of blood interferon genes between sex groups. Boxes represent the mean gene expression value for women (**left**) and men (**right**) groups at blood samples. Old is defined as >60 years old. Box line color indicates the significance level, red: *p* < 0.05 and fold change (FC) > 0.5; orange: *p* < 0.05 and FC > 0.5 and blue: *p* > 0.05 (not significant). The model indicates that females present a more activated antiviral response than males for antiviral interferons, B-cell homing and cell survival pathways. Created with biorender.com.

**Figure 6 genes-11-01447-f006:**
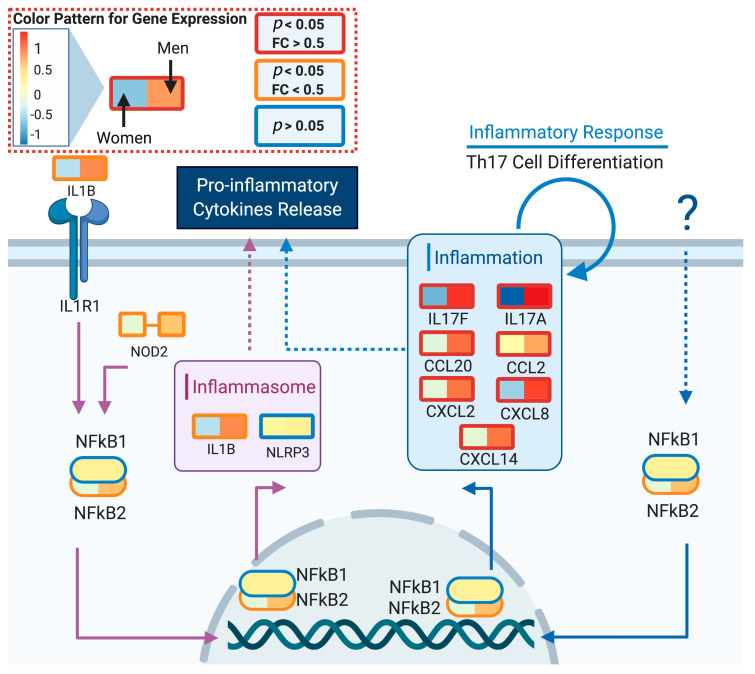
Differential expression of spleen cytokine and chemokine genes between sex groups. Boxes represent the mean gene expression value for old women (**left**) and men (**right**) groups at spleen samples. Old is defined as >60 years old. Box line color indicates the significance level, red: *p* < 0.05 and fold change (FC) > 0.5; orange: *p* < 0.05 and FC > 0.5 and blue: *p* > 0.05 (not significant). The model indicates that males have higher basal levels of proinflammatory cytokines. Created with biorender.com.

**Figure 7 genes-11-01447-f007:**
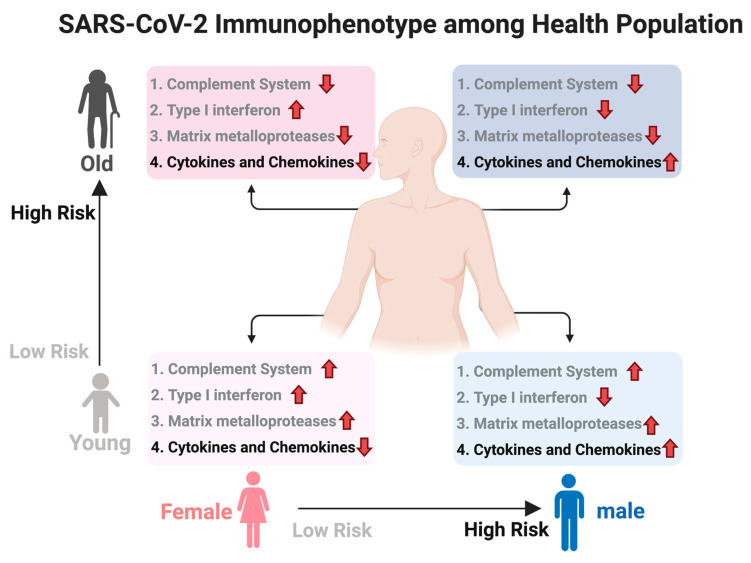
Summary of immunophenotype differences among demographic groups relevant to COVID-19. Different groups have a distinct basal immunophenotype that may contribute to their different clinical outcomes when infected with SARS-CoV-2. Here old is considered an age higher than 60 y. The direction of the arrow indicates relative higher (up) or lower (down) expression for that group of genes. Groups 1–3 have a protective role while group 4 associated to higher susceptibility. Created with biorender.com/.

**Table 1 genes-11-01447-t001:** Number of differentially expressed COVID-19 Disease Map genes and COVID-19 related pathways genes between sex and age groups in different human tissues and cell types. The number of significant (*p*-value (*p*) < 0.05) genes are reported either with or without a fold-change (FC) cutoff value of 0.5.

Analysis	Whole Blood	Kidney-Cortex	Stomach	Lung	Heart Left-Ventricle	Brain-Cortex	Spleen	Lymphocytes
Sex: *p* < 0.05; FC > 0.5	77	93	56	47	57	47	91	65
Sex: *p* < 0.05	438	184	213	179	252	270	480	193
Age: *p* < 0.05; FC > 0.5	104	69	76	41	51	45	99	104
Age: *p* < 0.05	330	184	408	180	183	799	515	315

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
