# Peer review of "Transcriptional Differences for COVID-19 Disease Map Genes between Males and Females Indicate a Different Basal Immunophenotype Relevant to the Disease"

_genes, 2020, doi:10.3390/genes11121447_

Round 1
Reviewer 1 Report
In the study “Transcriptional differences for COVID-19 Disease Map genes between males and females indicate a different basal immunophenotype relevant to the disease”, the authors utilize a valuable RNA-seq data as part of the GTEx project along with the COVID-19 Disease Map to easily search genes that are related to SARS-CoV-2-host interaction.
The study provides a nice tool for other researchers to utilize for further evaluating COVID-19 related gene expression in a large organ-specific manner amongst healthy patients. The authors utilize this tool to evaluate differences in immune phenotypes between different sexes that has been extensively investigated previously. The authors note that there are differences in gender and age risk factors amongst other coronaviruses implying SARS-CoV-2 regulation of the immune system accounting for risks associated with older age and male gender. However, this study is evaluating fundamental differences in immune system amongst healthy, non-infected adults and it is unclear how the basal immunophenotype differences would generate such differential risk factors related to age and gender for different coronaviruses. As noted by the authors, other reports have evaluated immune system differences with gender in SARS-CoV-2 infected patients, however this appears to uniquely look at fundamental immune differences between genders that could have an impact.
The group discusses age related effects on the immune system in Figure 1 and 3 however the impact of age is never discussed in the discussion. I would either remove the impact of age in the figures or change the title and include discussion of age on immune system in the Discussion.
Minor comments:
- Figure 3b – the multiple colors for each gene is difficult to follow. The genes on the left which seem to be elevated in females involved in anti-viral and on the right in males that are involved in inflammatory response. How are ages relating to this and represented in the barcode colors?
- Supplementary Figure 1 – Different is misspelled.
- Supplementary Figure 2 is not referred to in the paper and unclear what it adds to the paper.
Author Response
Major comments:
1) The group discusses age related effects on the immune system in Figure 1 and 3 however the impact of age is never discussed in the discussion. I would either remove the impact of age in the figures or change the title and include discussion of age on immune system in the Discussion.
Reply: Thank you for pointing out our shortcomings in the analysis. We have now substantially reshaped the paper to provide a more systematic analysis of both and gender. This has affected multiple sections of the manuscript. We found shared and specific groups of genes being differentially expressed associated to demographic groups at lower risk. These results are now represented in an extended Heatmap in figure 2 of the paper. We have also included a summarizing figure in the discussion. Hope this addresses the reviewer´s concerns
Minor comments:
1) Figure 3b – the multiple colors for each gene is difficult to follow. The genes on the left which seem to be elevated in females involved in anti-viral and on the right in males that are involved in inflammatory response. How are ages relating to this and represented in the barcode colors?
Reply: Sorry for the confusing the figure. We thought it was important to show the different expression levels across sexes (now also age) levels for genes showed in the figure to stress the message of the molecular differences. However, we understand that there were too many colors at each gene box. We have now re-edited figures to convey a clearer message and only indicated a different color for one of the comparisons, either sex or age.
2)Supplementary Figure 1 – Different is misspelled.
Reply: Thank you. We have deleted Supplementary figure 1.
3)Supplementary Figure 2 is not referred to in the paper and unclear what it adds to the paper.
Reply: Sorry for not being clear with this Supplementary Figure. Supplementary Figure 2 support our decision of removing the ventilator case samples from the blood tissue for inclusion in the DeCovid app. Although this is not further used in the rest of the paper, we believe it is important information to justify our choice. This is now Supplementary figure 1.
Minor comments:
1) As minor criticisms, there are a number of points where the language is not clear leading to confusion e.g. "However, which specific immunological characteristics imprint an existing condition that how this relates to COVID-19 remains to be explored."
Reply: Thank you for pointing it out. We have revised the English for clarity. The indicated sentence has been re-written to: However, which specific aspects of the sex and age related immunophenotype represent an existing condition and how this relates to COVID-19 remains to be explored.
2) Sup. Fig. 1 appears to just be data collected from another source and is not really relevant to the paper.
Reply: We have deleted Sup. Fig. 1. Thank you for your advice.

Reviewer 2 Report
In the paper, the authors present the DeCovid app, a shiny app for use with R, or use with docker. This app is relatively easy to install and use and may be of use to researchers investigating gene expression levels. Within the app the results are mostly very clearly presented. The integration with Gene Ontologies is also interesting and a useful tool for interpreting results. There are some small errors with the language in the interactive parts of the app which should be corrected to encourage wider use. While the app works well, it is limited by essentially having two controlling factors (age & gender). If the user could specify control groups this would become a more powerful and useful tool.
The DeCovid app has been used to generate results which are then presented within the paper as the original research contribution of the paper. In the results/discussion sections the results are presented with sufficient context however no results are discussed in sufficient depth or with insight into their significance. Datasets from several tissues are presented for use with DeCovid but only one dataset is analysed. This seems like a missed opportunity for some comparative insights. Age is used as a controlling factor, but at what banding does the difference become significant? Is there a difference between different age ranges, rather than just old/young? I'm also not sure that the PCA in Sup. Fig. 2 is as clear as suggested, and it would be interesting to see how results change based on this data. The results presented are not analysed as a whole, but rather just a point-by-point discussion.
The merit of the paper is in the utility of the DeCovid app. I think with a more robust framework for control variables, this would be a useful tool and suitable for publication in a journal focussed on bioinformatics resources.
As minor criticisms, there are a number of points where the language is not clear leading to confusion e.g. "However, which specific immunological characteristics imprint an existing condition that how this relates to COVID-19 remains to be explored."
Sup. Fig. 1 appears to just be data collected from another source and is not really relevant to the paper.
Author Response
------------------------Reviewer 2------------------------
Major comments:
1) While the app works well, it is limited by essentially having two controlling factors (age & gender). If the user could specify control groups this would become a more powerful and useful tool.
Reply: Thank you for giving great advice to make the app better. We add an advanced analysis part in our new version of the app that allows the selection of specific groups as experimental or control groups. You can download our new version of the app at https://github.com/ConesaLab/DeCovid.
2) In the results/discussion sections the results are presented with sufficient context however no results are discussed in sufficient depth or with insight into their significance. Datasets from several tissues are presented for use with DeCovid but only one dataset is analyzed. This seems like a missed opportunity for some comparative insights.
Reply: Thank you for agreeing our results are sufficient and pointing out our shortage in the analysis. In the revision, we analyzed different and critical immune tissues/cells (blood, spleen, and lymphocytes)to find out baseline immune pattern differences in both age and sex. Please see Figure 2 with a heatmap showing these results. We have included a more extended discussion of the significance of these results and analyzed similarities and differences between the gene expression signatures of demographic groups that are at lower risk (females and young). We have also included a last figure in the paper to summarize these findings.
3) Age is used as a controlling factor, but at what banding does the difference become significant? Is there a difference between different age ranges, rather than just old/young?
Reply: Sorry for leading you confused about it. In the paper, we selected all the samples with an age higher than 60 as old people. We now provide an analysis of differences in expression between age groups and incorporate in the app the option to modify the age threshold for determining age groups.
4) I'm also not sure that the PCA in Sup. Fig. 2 is as clear as suggested, and it would be interesting to see how results change based on this data.
Reply: Sorry to lead your uncertainty. Possibly our previous PCA plot is not clear enough. We created a new PCA plot using the same data with the value scaled in-unit variance, which helps to see the contribution of another covariate (not simply the PC1 but also PC2). We can find the bias early from this plot since all the ventilator cases clustered together and separated from other cases. Besides, we also analyzed the result with the ventilator case. The result changes totally due to the ventilator case's bias, and hard for us to find the interferons and cytokines pattern when sex is an experimental factor. Also, we cannot find a higher expression of complement system genes in young people when age is an experimental factor due to the bias. Therefore, we believe removing the ventilator cases in the blood sample is essential.
5) The results presented are not analyzed as a whole, but rather just a point-by-point discussion.
Reply: Thank you for pointing out our shortcomings. This time we analyzed the whole immune tissues/cells and found the common immune pattern across different groups to avoid a point-to-point discussion. We have now included a summarizing figure of the results of our analysis that show commonalities and specific gene expression patterns across demographic groups. We hope this contributes to providing a general view of how the expression of COVID-19 related genes may contribute to differences in disease severity among these groups.
6) The merit of the paper is in the utility of the DeCovid app. I think with a more robust framework for control variables, this would be a useful tool and suitable for publication in a journal focused on bioinformatics resources.
Reply: We are glad you like our DeCovid app and give suggestions for improvement. We followed your advice the app allows now users to customize control groups and experimental groups in different tissues/cells. Our research aims to characterize the underlying immunophenotype of different demographic groups that could lead to their different responses to SARS-CoV-2 infection. We have now improved both the presented analyses and the extent of the discussion, and we have also included more features in the app. We believe readers of the GENES journal will enjoy and appreciate this contribution.

Round 2
Reviewer 1 Report
I thank the authors for their consideration of the previous comments and additional data analysis evaluating the role of complement system and matrix metalloproteinase gene expression.
I found the new figures to be particularly helpful in going through the data and particularly the final summary figure.
For Figure 3: the left panel is delineated as epxrimental factor: sex and the right experimental factor: age. Would it be possible in this case to have two comparisons without complicating the additional variable? For the left panel would it be possible to combine all the females and all the males and compare those two groups?
Similarly, for Figures 4,5, and 6 the comparison groups are restricted. Figure 4 looks to compare Old versus young but restricts to mean while figure 5 and 6 looks to compare differences between sexes but restricts to older patients. Are these restrictions necessary in order to gain significance suggesting contributor effects of both age and sex to varying degrees for all the genes evaluated.
Author Response
For Figure 3: the left panel is delineated as epxrimental factor: sex and the right experimental factor: age. Would it be possible in this case to have two comparisons without complicating the additional variable? For the left panel would it be possible to combine all the females and all the males and compare those two groups?
Response: Respectfully, we prefer to maintain the breakdown by the additional demographic factor in these heatmaps as the expression pattern is not always uniform within one group. For example, complement system genes in spleen are highly expressed in young men but not in old men, and the average does not reflect these differences. By providing this breakdown the consistency of expression of groups of genes in each of the presented contrasts can be seen. However, he have included a new version of the heatmap with averaged values per demographic group as Supplementary Figure 2.
Similarly, for Figures 4,5, and 6 the comparison groups are restricted. Figure 4 looks to compare Old versus young but restricts to mean while figure 5 and 6 looks to compare differences between sexes but restricts to older patients. Are these restrictions necessary in order to gain significance suggesting contributor effects of both age and sex to varying degrees for all the genes evaluated.
Response: We have now included figures 4-6 with the average value per demographic group, which makes sense in these figures as the expression patterns are homogeneous within the indicated group.
Reviewer 2 Report
The authors have done a good job modifying the manuscript. It now contains a clearer, more detailed exploration of the analysis that can be be performed by the DeCovid app, which highlights the utility of the approach. I would recommend acceptance of the paper for publication subject to the following minor changes;
- The use of English could still be improved, and a spell check is needed "demographich".
- In the figures 4/5/6, the arrows pointing to Old Men, Young Men/Old Women should be reversed. Currently they look a little like they are describing a change in the gene expression, whereas I believe they are labelling the sides of the boxes. If the arrows were reversed, I believe this would be clearer.
Author Response
Thank you for the comment. The manuscript has been gone through a round of proof-reading by an English native speaker.